# Small-Cell Lung Cancer—An Update on Targeted and Immunotherapies

**DOI:** 10.3390/ijms24098129

**Published:** 2023-05-01

**Authors:** Nicholas McNamee, Ines Pires da Silva, Adnan Nagrial, Bo Gao

**Affiliations:** 1Blacktown and Westmead Hospitals, Sydney, NSW 2145, Australia; 2Westmead Clinical School, University of Sydney, Sydney, NSW 2145, Australia

**Keywords:** SCLC, immunotherapy, antibody–drug conjugate, PARP inhibitors, targeted therapy

## Abstract

Small-cell lung cancer (SCLC) is an aggressive disease with distinct biological and clinical features. The clinical course of SCLC is generally characterised by initial sensitivity to DNA-damaging therapies, followed by early relapse and broad cross resistance to second line agents. Whilst there has been an enormous expansion of effective targeted and immune-based therapeutic options for non-small cell lung cancer (NSCLC) in the last decade, little improvement has been achieved in SCLC treatment and survival due, at least in part, to underappreciated inter- and intra-tumoral heterogeneity. Here we review the current treatment paradigm of SCLC including recent advances made in utilizing immunotherapy and the challenges of identifying a predictive biomarker for immunotherapy response. We examine emerging new targeted therapies, combination immunotherapy and future directions of SCLC treatment research.

## 1. Introduction

Small cell lung cancer (SCLC) is a poorly differentiated neuroendocrine tumour which represents approximately 15% of all lung cancer diagnoses [1]. It is strongly associated with smoking and characterized by a predilection for rapid growth, early distant metastases and acquired drug resistance. Up to one in five have brain metastases at diagnosis [2]. The standard chemotherapy regimen for SCLC, consisting of a platinum agent combined with etoposide, was defined several decades ago [3]. Despite remarkable initial sensitivity to chemotherapy and radiotherapy, the response is usually short-lived [4]. The results of immunotherapy trials in SCLC, whilst positive, have been disappointing with only modest absolute improvements in survival when compared to outcomes in other tumour streams. The adding of the programmed death-ligand 1 (PDL1) inhibitor atezolizumab into platinum-based frontline chemotherapy of extensive stage SCLC (ES-SCLC) has improved median overall survival from 10.3 months to 12.3 months, with only 34.0% of patients remaining alive at 18 months [5]. This highlights the critical need for novel agents and improved therapies for this lethal disease. 

## 2. Pathological and Genomic Profiles of SCLC

SCLC is a highly proliferative epithelial neuroendocrine tumour [6]. A diagnosis of SCLC is made primarily by examination of haematoxylin and eosin-stained slides by light microscopy and observation of characteristic appearance of small round uniform cells with scant cytoplasm, ill-defined cell borders and distinctive nuclear features (fine granular chromatin lacking prominent nucleoli). The Ki-67 proliferation index is consistently high (50–100%). Typical immunohistochemistry results show expression of epithelial markers such as keratin and neuroendocrine markers including synaptophysin, chromogranin A and insulinoma-associated protein 1 (INSM1). 

There has been an increasing understanding of the molecular and genomic heterogeneity in SCLC as part of efforts to identify therapeutic vulnerabilities to improve outcomes. Key genomic profiling studies of SCLC, including comprehensive whole exome and whole genomic genome analyses were published in 2012 and 2015 [7,8]. SCLC genomes exhibited an extremely high mutation rate of 8.62 nonsynonymous mutations per million base pairs. There was nearly universal functional loss of two key tumour suppressor genes, *TP53* and *RB1*, sometimes by complex genomic rearrangements. Targetable mutations in known oncogenes, including *BRAF*, *PTEN*, *PIK3CA* and *SLIT2* were only found in rare cases which may provide a possible therapeutic opportunity of these individual patients. In contrast, a high frequency of mutations affecting known epigenetic regulators including histone-modifying genes and inactivating mutations in *Notch* family members were observed. These targets are a focus of intense ongoing investigation. 

Using gene expression data and non-negative matrix factorization, a novel approach to classification of subtypes of SCLC was published by Gay et al. in 2021 [9]. Four subtypes with distinct transcriptional characteristics and therapeutic vulnerabilities were defined largely by their differential expression of transcription factors *ASCL1*, *NEUROD1* and *POU2F3* (Table 1). The clinical implications of this subtype classification are significant, as each subtype demonstrates a unique vulnerability to investigational therapies. 

## 3. Current Standard Treatment

The American Joint Committee on Cancer (AJCC) and the International Association for the Study of Lung Cancer (IASLC) recommended the using of TNM staging system for SCLC as well as non-small cell lung cancer (NSCLC) [10]. In clinical practice, however, patients with SCLC are typically divided into limited versus extensive disease using the 2-stage Veteran’s Administration Lung Study Group (VALSG) staging system [11]. Limited stage SCLC (LS-SCLC) is disease confined to ipsilateral hemithorax and reginal lymph nodes that can be enclosed in one radiation field. ES-SCLC is more prevalent and includes malignant pleural or pericardial effusion alone, or any distant metastasis.

### 3.1. Limited Stage SCLC

For patients with LS-SCLC, the current standard treatment is concurrent thoracic radiotherapy and chemotherapy with cisplatin and etoposide. This is based on a meta-analysis of 13 trials that showed a 14% reduction in the mortality in patients treated with radiotherapy in addition to chemotherapy [12]. Cisplatin and etoposide was the preferred regimen for three decades [13], but carboplatin and etoposide has shown equivalent efficacy in a randomised trial with limited stage patients [14] and is commonly used in patients who have a contraindication to cisplatin, or patients who have a poor performance status.

A small fraction of patients limited stage disease (TNM stage I-IIA) may be candidates for up-front surgical resection. A retrospective analysis from the National Cancer Database in the United States showed, when matched based on stage, there was an improvement in median overall survival (OS) to 38 months compared to 22 months for these node-negative patients who underwent initial surgery [15]. Treatment recommendations are based only on retrospective data which are limited by unavoidable selection bias and information bias. There are no trials comparing alternative treatment regimens or combined modality treatment for these patients. A mediastinoscopy should be performed prior to resection to rule out occult nodal disease. Following surgery, adjuvant systemic therapy with cisplatin and etoposide has shown an improved median OS of 66.0 months compared to 42.1 months if no chemotherapy in a large retrospective analysis [16]. If nodal involvement is found following surgery, thoracic radiotherapy is recommended, although there is limited evidence to guide treatment in this pathologically upstaged group [17,18]. 

### 3.2. Extensive Stage SCLC

About 85% of patients with SCLC have extensive stage disease at the time of diagnosis [19]. Until recently, the standard of care treatment had been platinum-based chemotherapy. Outcomes of chemotherapy alone are poor, with a median OS of 9.4 months despite an overall response rate (ORR) of 66.0% based on the COCIS meta-analysis [20]. 

The addition of immune checkpoint inhibitors to this treatment has been the only improvement in SCLC for many years that has improved survival, as discussed below, and summarised in Table 2. The IMpower133 trial, published in 2018, was the first randomised trial to demonstrate progression-free survival (PFS) or OS improvement with immune checkpoint blockade (ICB) in first line treatment in ES-SCLC [5]. This trial added atezolizumab to standard of care platinum-based chemotherapy, in this case carboplatin and etoposide. The improvement of median OS of 2 months (12.3 months vs. 10.3 months; *p* = 0.007) was celebrated as the most clinically meaningful survival prolongation in SCLC for decades. 

The CASPIAN trial published in 2019 was a randomised (1:1:1), open-label, three-arm trial assessing PDL1 blockade with durvalumab, with or without the CTLA4 inhibitor tremelimumab, in combination with platinum-etoposide (EP) in treatment-naïve ES-SCLC [21,22]. Consistent with findings from IMpower133, adding durvalumab to platinum-etoposide showed a significant improvement of OS to 12.9 months versus 10.5 months (*p* = 0.032) in platinum-etoposide only patients. However, the addition of tremelimumab to durvalumab plus EP did not significantly improve outcome versus EP. This was the first report of a phase III trial evaluating dual ICB in combination with chemotherapy in ES-SCLC. The increased toxicity observed with the addition of a CTLA4 inhibitor resulting in lower total exposure to durvalumab could have contributed to the absence of significant benefit compared with platinum-etoposide. 

KEYNOTE-604 was published in 2020 and reported on pembrolizumab in combination with platinum-etoposide also as first line treatment for ES-SCLC [23]. The addition of pembrolizumab to EP significantly improved PFS (HR = 0.75; *p* = 0.0023). Although the OS hazard ratio favoured pembrolizumab plus EP, the prespecified significance threshold of *p* = 0.0128 was narrowly missed (HR = 0.80; *p* = 0.164). The lack of a statistically significant OS benefit was unexpected based on the results of CASPIAN and IMpower133. The shorter than expected OS in KEYNOTE-604 may have been due to a high proportion of patients with brain metastases, larger tumour dimensions, and ≥3 metastases at baseline. Given this negative result, pembrolizumab is not approved for use in first line ES-SCLC. 

CheckMate 451, published in 2021, examined the potential benefit of maintenance nivolumab and ipilimumab for four cycles followed by nivolumab alone, or nivolumab alone after chemotherapy or placebo in patients who had responded the first line platinum-based chemotherapy [24]. There was no statistically significant difference in OS for either immunotherapy regimen compared to placebo, as summarised in Table 2. As might be expected, the combination of PDL1 inhibitor and CTLA4 inhibitor arm had significantly higher grade 3–4 toxicity of 52.2% compared to 11.5% for nivolumab alone and 8.4% for placebo. This may have contributed to reduced drug exposure in that arm and the negative result. 

Published in 2022, CAPSTONE-1 examined the novel PD-L1 inhibitor adebrelimab with EP chemotherapy [25]. This study of 462 patients was positive and reported a mOS of 15.3 mo in the treatment group compared to 12.8 mo for placebo (*p* = 0.0017, HR 0.72). In contrast to KEYNOTE-604, there were just five patients in each group with brain metastases which may go some way to explaining the improved OS in both the intervention and placebo arms. As identified by the authors, there was also a high proportion of patients who received subsequent systemic therapies beyond the clinical trial compared to the trials discussed above. Despite these limitations, as well as the restriction of this study to only include patients from China, it is clearly supportive of the addition of ICB to EP for ES-SCLC. 

The most recent phase III study of ICB in addition to chemotherapy is ASTRUM-005 published in September 2022 which included 585 patients treated with EP +/− serplulimab, a fully humanised IgG4 monoclonal antibody against the PD-1 receptor [26]. The immunotherapy group reported mOS of 15.4 mo compared to 10.9 for EP alone (*p* < 0.001). Of note, 19.8% of patients in the trial were never smokers, a proportion higher than previous multinational trials with larger fractions of non-Asian patients. An improvement in OS was seen in this unique patient subgroup, consistent with that in the overall population. 

Taking these pivotal immunotherapy trials together, there is clearly benefit from the addition of checkpoint inhibitors to platinum-based chemotherapy. As has been the case in other tumour types, the long tails of the Kaplan–Meier curves in these trials suggest that there is a small subset of patients, whom we are still ill-equipped to identify, that may have deep and durable responses to ICB. Unfortunately, there has not been any reliable or robust prognostic biomarkers identified. PD-L1 expression was not predictive of efficacy in these trials, and likewise tumour mutational burden (TMB) is of limited value [21,25,27].

**Table 2 ijms-24-08129-t002:** Overview of checkpoint inhibitor trials in ES-SCLC.

Study	Year	Agents	Phase; Line (n)	Key Results
First Line Treatment				
IMpower133 [5]	2018	EP +/− atezolizumab	III; 1 (403)	EP + atezolizumab: mOS 12.3 mo EP + placebo: mOS 10.3 mo (*p* = 0.0154)
CASPIAN [22]	2019	EP + durvalumab + tremelimumab or EP + durvalumab or EP alone	III; 1 (805)	EP + durva + trem: mOS 10.4 mo EP + durva: mOS 12.9 mo EP alone: mOS 10.5 mo
KEYNOTE-604 [23]	2020	EP +/− pembrolizumab	III; 1 (453)	EP + pembrolizumab: mOS 10.8 mo EP + placebo: 9.7 mo (*p* = 0.0164 ^1^)
CheckMate 451 [24]	2021	EP -> ipilimumab + nivolumab followed by nivolumab, or EP -> nivolumab, or EP -> placebo	III; 1 (849)	EP -> ipi/nivo: mOS 9.2 mo EP -> nivo: mOS 10.4 mo EP -> placebo: 9.6 mo
CAPSTONE-1 [25]	2022	EP +/− adebrelimab	III; 1 (462)	EP + adebrelimab: mOS 15.3 mo EP + placebo: mOS 12.8 mo (*p* = 0.0017)
ASTRUM-005 [26]	2022	EP +/− serplulimab	III; 1 (585)	EP + serplulimab: mOS 15.4 mo EP + placebo: mOS 10.9 mo (*p* < 0.001)
Recurrent Disease				
CheckMate 032 [28]	2016	Nivolumab ^2^	I/II; ≥3 (109)	mPFS 1.4 mo; mOS 5.6 mo
KEYNOTE-028, KEYNOTE-158 [29]	2020 ^3^	Pembrolizumab (10 mg/kg q2 weeks, n = 19, 200 mg q3 weeks n = 64)	Ib/II; ≥2 (83)	mPFS 2.0 mo, mOS 7.7 mo (*mOS 14.6 mo for PD-L1 CPS ≥ 1)*

EP: platinum-etoposide chemotherapy, OS: overall survival, PFS: progression free survival, CPS: combined positive score; ^1^ Did not reach pre-specified significance boundary; ^2^ Pooled data for nivolumab monotherapy—study also included combination nivolumab/ipilimumab arms; ^3^ Retrospective pooled analysis of both multicohort studies (KEYNOTE-028, n = 19; KEYNOTE-158, n = 64).

## 4. Recurrent Disease

Despite rapid and impressive initial response to first line platinum-etoposide with or without immunotherapy, virtually all patients eventually relapse. For recurrent disease, chemotherapy outcomes are poor and immunotherapy trials’ results have been disappointing, as summarised in Table 2. Although topotecan is not the most desirable drug in the second line setting because of its toxicity profile, it has been difficult for new agents to “beat” this drug [30,31]. Table 3 summarises the approved drugs for ES-SCLC.

### 4.1. Chemotherapy

Topotecan is the standard second line treatment for ES-SCLC based on a phase III trial published in 1999 which showed equivalent efficiency (PFS 13.3 vs. 12.3 weeks; *p* = 0.552) but improved symptom control. This was by comparison with the more toxic regimen of CAV (doxorubicin, cyclophosphamide, vincristine) [33,34]. 

Patients who experience progression ≥ 90 days from the completion of first-line chemotherapy are generally considered as platinum sensitive. A comparison of rechallenge with platinum-based chemotherapy compared to second line topotecan was published in 2020 involving 164 platinum sensitive recurrent SCLC patients in France [35]. PFS was significantly longer in the platinum chemotherapy group being 4.7 months compared to 2.7 months in the topotecan group. The platinum group had a response rate of 49% compared to 25% in the topotecan group. There was not, however, a statistically significant difference in OS at time of analysis. 

In patients not fit enough to consider any intravenous chemotherapy, oral topotecan has shown an ORR of 7% and increased survival from 13.9 weeks to 25.9 weeks with improved symptom control compared to best supportive care in a small, randomised study of 141 patients in 2006 [33]. Other single agents can be used sequentially, with diminishing response rates and duration of response, with very few patients achieving disease control after the third line of treatment [36]. 

### 4.2. Immunotherapy

Like all emerging therapies, ICB was first tested in patients recurrent ES-SCLC who had failed standard treatment. The modest efficacy observed led to the rapid launching of trials in the first line setting a few years later, as discussed in Section 3.2. On the basis of a 2-year OS rate of approximately 20% reported in CheckMate 032 [28] and in the pooled analysis from the phase Ib KEYNOTE-028 [29] and phase II KEYNOTE-158 [37] trials, the Food and Drug Administration (FDA), but not the European Medicines Agency (EMA), initially approved nivolumab in 2018 and pembrolizumab in 2019 as third-line treatments in patients with ES-SCLC. 

However, the phase III confirmatory studies did not show any statistically significant survival benefit when compared to chemotherapy in the second line setting with nivolumab (CheckMate 331) [30]. The indication has since been withdrawn for nivolumab, although pembrolizumab remains an approved option after at least 2 prior lines of therapy, which must include platinum chemotherapy [38]. Notably, the CheckMate 032, KEYNOTE-028 and KEYNOTE-158 clinical trials excluded patients who had previously received immune checkpoint blockade. 

### 4.3. Alkylating Agent—Lurbinectedin

Lurbinectedin is an alkylating drug and an analogue of tetrahydroisoquinoline trabectedin which induces degradation of transcribing RNA Pol II causing DNA damage with demonstrated antitumour activity in animal models [39]. In a single-arm phase II trial of 105 patients, referred to as Study B-005, lurbinectedin elicited an ORR of 35% and duration of response of 5.3 months with an acceptable and manageable safety profile [40]. Efficacy was observed in both platinum-sensitive and resistant populations with ORR of 45% and 22%, respectively [41]. It received FDA accelerated approval as second line treatment metastatic SCLC in June 2020. Continued approval is contingent upon verification of clinical benefit in a confirmatory trial, LAGOON (NCT05153239) which compares lurbinectedin as single-agent or in combination with irinotecan versus topotecan or irinotecan alone. Separate to this, ATLANTIS (NCT02566993) is a randomised phase III trial of lurbinectedin in combination with doxorubicin as a second-line treatment for ES-SCLC, as compared with physician’s choice of chemotherapy, either topotecan or CAV. The primary end point is OS [39]. 

## 5. Emerging Therapies

### 5.1. Antibody–Drug Conjugates (ADCs)

Compared to classic chemotherapy, antibody–drug conjugates (ADCs) are designed to achieve a wider therapeutic window with minimal off-target side effects. 

All ADCs have three main components: an antibody targeted against a tumour-associated antigen, a linker, and a cytotoxic payload. The ideal antibody target is a cell surface protein that is overexpressed by tumour cells but not on normal cells thus allowing more selective killing. The linker binds the antibody to the cytotoxic agents and is usually either of cleavable (hydrazone, disulfide, and dipeptide) or non-cleavable varieties. Several cytotoxic payloads have been used ranging from microtubule inhibitors (auristatin) to DNA cleavage agents (calicheamicin) and topoisomerase inhibitors (camptothecin). 

#### 5.1.1. DLL-3 Targeting ADCs

Delta-like ligand 3 (DLL3) is a NOTCH ligand involved in regulating neuroendocrine differentiation which is expressed in up to 75% of SCLC, whilst exhibiting minimal to absent surface expression in normal tissues [42]. Notch pathway deregulation is a crucial event in SCLC tumorigenesis, disease progression and chemoresistance and its inhibition has been shown to effective in vivo animal models [43,44]. This therefore represents an appealing novel biomarker and a potential target in SCLC. 

Rovalpituzumab tesirine (Rova-T, SC16LD6.5) is the first DLL3-targeted antibody tethered to a cytotoxic payload (pyrrolobenzodiazepine) by means of a protease-cleavable linker. The first-in-human phase I trial published in 2017 elicited significant enthusiasm because of the efficacy results in ES-SCLC- an ORR rate of 18% and an ORR of 38% in DLL3-high (>50% expression) ES-SCLC [45]. In addition to the striking efficacy, reported toxicity was minimal in this trial. These promising early results prompted the rapid launching of several phase I, II and III clinical trials, testing Rova-T in different settings in patients with extensive SCLC.

Four independent studies were published in the Journal of Thoracic Oncology in 2021, namely MERU (Rova-T versus placebo as a maintenance treatment in patients with SCLC after platinum-based chemotherapy regardless of DLL3 expression) [46], TAHOE (Rova-T versus topotecan as second-line therapy in high DLL3 expression SCLC) [47], and two phase I/II trials [48,49]. The results demonstrated that Rova-T is not effective against SCLC, casting a pall over the future of the therapy and closing a door that seemed opened four years ago. 

Rova-T has been evaluated in combination with budigalimab [38] or nivolumab with or without ipilimumab [48] for previously treated ES-SCLC. Although the response rates were encouraging, the combination of Rova-T with ICB had high rates of toxicities. For example, the triple combination arm reported 92% of patients experienced grade ≥ 3 adverse events, compared to 53% in the Rova-T and nivolumab alone group. 

#### 5.1.2. TROP-2 Directed ADCs

Trophoblastic cell-surface antigen-2 (TROP-2) is a transmembrane calcium signal transducer that is overexpressed in many epithelia cancers including triple-negative breast cancer (TNBC), NSCLC, and SCLC [50].

Sacituzumab govitecan is a TROP-2 directed antibody bound to SN-38, the active metabolite of irinotecan and a topoisomerase I inhibitor, through a hydrolysable linker (Figure 1). Based on the results of studies IMMU-132-05 and IMMU-13 [51], it has been approved in certain countries for the treatment of patients with metastatic TNBC [52] and metastatic urothelial cancer [53]. It has been evaluated in patients with SCLC in the IMMU-132-01 study and the ORR based on local response was 17.7% and median duration of response was 5.7 months (range 3.6 to 19.9 months) [51]. A further expansion cohort is being recruited into IMMU-132-11 phase II study to assess its activity in ES-SCLC. 

### 5.2. PARP Inhibitors

While SCLC is not characterised by *BRCA* mutations or homologous recombination deficiency, aberrant expression of several genes implicated in DNA damage repair is a common observation. Nearly 100% of cases of SCLC have homozygous loss or inactivation of *RB1*, encoding the primary regulator of the G1-S cell cycle checkpoint and *TP53*, critical for multiple DNA damage response pathways [7]. The notable sensitivity of SCLC to DNA damage, including DNA damaging agents such as covalent DNA adducts and crosslinks (platinum, temozolomide) or single-strand or double-strand breaks (etoposide, topotecan, irinotecan), suggests that targeted inhibition of DNA pathways may be a particularly attractive strategy. 

Poly (ADP-ribose) polymerase (PARP) is an essential nuclear enzyme for the repair of single-stranded DNA (ssDNA) breaks through the base excision repair (BER) pathways [54]. PARP inhibitors function by blocking PARP mediated DNA repair and is expected to sensitize tumour cells to cytotoxic agents which induce DNA damage. PARP inhibitors have shown some activity in SCLC preclinical models [55]. However, the single agent activity of PARP inhibitors in SCLC appears to be minimal. Notably, the UK STOMP trial failed to show an improvement in PFS for patients treated with maintenance olaparib after the completion of first line platinum-based chemotherapy [56]. 

PARP inhibitors are being extensively explored in various combinations for treating SCLC [57]. Combination studies with available results are summarised in Table 4. The addition of veliparib to temozolomide (TMZ), an oral alkylating agent with single agent activity in SCLC, showed improved ORR from 14% to 39% in a placebo-controlled randomised phase II trial, but no significant difference in 4-month PFS was noted [58]. Significantly prolonged PFS (5.7 vs. 3.6 months; *p* = 0.009) and OS (12.2 vs. 7.5 months; *p* = 0.014) were observed in patients with SLFN11-positive tumours treated with the TMZ and veliparib combination, as further discussed below. A combination of TMZ and olaparib has been studied in a phase II study of 50 patients with recurrent, previously treated SCLC [59]. This study had an OS of 8.5 months and overall response rate of 41.7%. In these trials, TMZ was often administered at doses of 75 mg/m^2^ daily for seven days per 21-day cycle. There is evidence that lower continuous dose TMZ can potentiate the cytotoxic activity of PARP inhibitors by forming PARP-DNA-trapping complexes to induce cytotoxicity, proposing an alternative strategy of combing PARP inhibitors with lower dose, continuous TMZ. IMP4297-106 is a phase Ib/II open label study to assess the efficacy of senaparib with TMZ at 30mg flat dose daily, day 1–21 out of a 28-day cycle in patients with advanced solid tumours with a planned expansion cohort in ES-SCLC after first line platinum-based chemotherapy [60].

Combination of PARP inhibitors with immune checkpoint inhibitors have had mixed results. Treatment of SCLC with olaparib led to an upregulation of PD-L1 expression in in vitro models and subsequent increased sensitivity to combination treatment [61]. In a phase II trial of 20 patients with refractory SCLC treated with durvalumab an olaparib was an overall response rate of only 10.5% [62]. 

**Table 4 ijms-24-08129-t004:** Published PARP inhibitor combination clinical trials in recurrent ES-SCLC.

Agents	Study Design	Key Results
Veliparib + temozolomide [58]	Phase II, randomized, placebo-controlled N = 104	TMZ/veliparib mOS 8.2 mo; ORR 39% TMZ/placebo mOS 7.0 mo; ORR 14% SLFN11 predicted improved PARPi response
Olaparib + temozolomide [59]	Phase I/II, single-arm N = 48	mOS 6.7 mo; ORR 41% *
Fuzuloparib + SHR-1316 (PD-L1 inhibitor) [63]	Phase Ib, multi-stage N = 23	mOS 5.6 mo; ORR 6.3% *
Olaparib + durvalumab [62]	Phase II, single-arm N = 20	mOS 4.1 mo; ORR 10.5% *1 CR in BRCA-mutant disease*

* Results for patients treated with RP2D; PARPi: PARP inhibitor; CR: complete response.

SLFN11 (schlafen family member 11) has been identified as a novel predictive biomarker of response to talazoparib [64]. In the same study, Lok et al. also demonstrated in vitro, using CRISPR gene editing techniques, that loss of SLFN11 was associated with resistance to talazoparib. This finding has been further explored and supported in patients with recurrent SCLC in a phase II study of temozolomide with or without veliparib. One hundred and four patients were included and whilst there was no statistically significant difference between the treatment and placebo groups, a subgroup analysis stratified on SLFN11 expression showed improved PFS and OS in patients receiving temozolomide and veliparib whose tumours express SLFN11 [58]. The combination with temozolomide was based on previous data showing activity in refractory SCLC as well as synergistic activity in vivo with PARP inhibitors [64,65]. 

### 5.3. Anti-Angiogenesis

A 2021 meta-analysis examined the role of bevacizumab with chemotherapy in first-line ES-SCLC [66]. This included 368 first line patients, and 108 relapsed patients across nine studies. There was a statistically significant improvement in PFS with HR 0.74 (*p* = 0.007); however, no statistically significant survival benefit. For relapsed patients, there was an ORR of 19% compared to 71% for the untreated patients. A 2017 meta-analysis of 1322 patients who received any anti-angiogenesis agents in randomised controlled trials similarly concluded that there was no improvement to PFS, OS, or ORR in combination with chemotherapy [67]. 

Table 5 summarises the available data for oral anti-angiogenic agents in recurrent ES-SCLC. Sunitinib maintenance after chemotherapy as part of first line treatment of ES-SCLC was studied in a phase II randomised trial in 2015 [68]. This trial randomised 95 patients to maintenance sunitinib, of who 41 received placebo and 44 received sunitinib. Whilst there was an improvement in median PFS of 3.7 months compared to 2.1 months (*p* = 0.02), there was no statistically significant improvement in median OS (6.9 months for placebo, 9.0 months for sunitinib, *p* = 0.16). 

Anti-angiogenesis agents, namely bevacizumab, sorafenib and cediranib, have so far not had a meaningful impact as monotherapy for recurrent SCLC. Cediranib was examined in a phase II study of 25 patients who had progressed after platinum-based chemotherapy [69]. This study demonstrated no confirmed objective responses at either a 45 mg or 40 mg dose of cediranib, and had poor mPFS and mOS, respectively. 

Sorafenib was trialled on 89 patients with recurrent disease in a phase II SWOG trial [70]. This trial showed an ORR of 11% using sorafenib 400 mg twice daily continuously and mOS of 5.3 months and 6.7 months for the platinum resistant and sensitive groups, respectively. Apatinib, a selective VEGF (vascular endothelial growth factor) receptor-2 inhibitor, has been examined in a single-arm study of 57 patients with recurrent ES-SCLC [71]. Encouragingly, this study demonstrated a median OS of 11.2 months in this pre-treated group treated with apatinib monotherapy. 

Anlotinib monotherapy, an antioangiogenic multi-kinase inhibitor, has been retrospectively studied in refractory SCLC patients who failed 2 prior lines of treatment [72]. Whilst this is a retrospective study, it reports a median OS of 7.8 months. A phase II trial of pazopanib also yielded disappointing results. In a study of 58 patients, the median OS was 6.0 months [73]. Results were slightly better when taking the platinum-sensitive group alone, with a median OS of 8.0 months. This study included circulating tumour cell (CTC) analysis, and the number of CTCs significantly correlated with treatment efficacy.

### 5.4. Tyrosine Kinase Inhibitors

C-kit expression has been reported in up to 88% of SCLC samples and high levels of expression have correlated with improved OS in retrospective data [74]. Despite these high levels of expression, imatinib has been demonstrated to have no significant antitumour activity [75]. 

The transformation of NSCLC to SCLC has been identified as a cause of EGFR-targeting TKI resistance. This mainly occurs in the light smoker and histologically combined subtypes [76]. Despite some patients maintaining the targetable mutation, EGFR-TKIs seem to have little benefit in small cell lung cancer [77,78]. 

Other targetable mutations in known oncogenes, including *BRAF*, *PTEN*, *PIK3CA,* and *SLIT2* are rare and their therapeutic value in SCLC is unknown. PIK3CA mutations were identified in 5 out of 28 SCLC samples [79]. PIK3CA inhibition has subsequently been shown to have some effect in pre-clinical models and has raised the potential of PIK3CA targeting as a therapeutic strategy [80].

### 5.5. Bispecific T-Cell Engagers

Bispecific T-cell engagers have been shown to be effective in haematological malignancies [81] but the poor distribution and short half-life of the macromolecules has so far restricted their application for solid tumours [82]. Tarlatamab (formerly AMG 757) is a novel DLL3-targeted bispecific T-cell engager for which a phase I study in refractory/recurrent ES-SCLC was published in January 2023 [83]. One hundred and seven patients received the investigational agent, of which 36% had brain metastases and 72% had received ≥2 prior lines of therapy. Notable toxicities were cytokine release syndrome (CRS) in 52%, although only 1% grade ≥3, as well as pyrexia in 40.2%. The mPFS was 3.7 mo and, impressively, mOS 13.2mo. Importantly, a post hoc analysis suggests that, retrospectively tested, elevated DLL3 expression correlates with improved responses. However, this is limited by the retrospective nature and low numbers in this study. 

Targeting the same DLL3 protein, AMG 119 is a chimeric antigen receptor (CAR) T cell therapy currently in a phase I trial [84] with expected completion in 2025 [85]. We eagerly await these results.

## 6. Discussion

SCLC has been a challenge for the era of personalised therapy due, at least in part, to underappreciated inter- and intra-tumour heterogeneity. In contrast to NSCLC, where biomarker selection for targeted and immune therapies has dramatically altered treatment approaches, clinical trials for SCLC have largely focused on unselected populations and have had predictably disappointing results in most cases. 

SCLC has one of the highest mutation loads with about 200 non-synonymous mutations per tumour. Conventional thought is that it should therefore be sensitive to immunotherapy, which has not been seen in trials thus far. The potential cause of this poor efficacy comprises a range of factors adverse to immunotherapy, such as low number of tumour infiltrating lymphocytes, low expressions of PD-L1 and MHC class, presence of immunosuppressive cell populations and cytokines, as well as avascular tumour areas from rapid tumour growth for immune evasion [86]. While ICB is now a standard-of-care for SCLC, predictive biomarkers for this therapeutic class have remained elusive, with evidence supporting (and opposing) TMB and PD-L1 expression. The stratification and analysis of survival outcomes based on molecular subtypes and genomic patterns may be a more nuanced and sophisticated approach which provides reliable prognostic information. For example, an exploratory analysis of survival based on molecular subtypes used the classification set out by Gay et al. above in 58 patients who had evaluable RNA data [87]. Limited by very small numbers, the inflamed subtype showed a trend towards higher median OS than the other three subtypes. However, this requires significant resources and local expertise to facilitate, and has been explored in only small numbers of patients. Future immunotherapy trials for SCLC should include pre-specified molecular analysis using a standardised and available tool to create data which are applicable to clinical practice. 

Maintenance durvalumab as consolidation therapy in patients with locally advanced, unresectable NSCLC without disease progression after two or more cycles of platinum-based definitive chemoradiation therapy has shown significant OS benefit (HR = 0.68; *p* = 0.0025) in the PACIFIC trial [88]. The ADRIATIC study is a phase III trial examining the benefit of maintenance durvalumab with or without tremelimumab compared to placebo as maintenance therapy after concurrent chemoradiation for LS-SCLC [89]. Completion is expected by late 2024. Similarly, the NRG Oncology/Alliance LU005 trial is currently recruiting for an open label study of chemoradiation with or without atezolizumab maintenance [90]. 

To advance immunotherapy of SCLC, we need to understand the significance of the high TMB and the influence of tumour microenvironment, tackle the obstacles of modest response rates and survival benefit either in combination or single agent therapy, lack of reliable biomarkers to predict response, the early emergence of resistance, and high treatment cost. Effective immunotherapy treatment of SCLC needs supplementary strategies to checkpoint inhibitors. Whether this is a vaccine in combination with ICB to overcome the resistance or other cytotoxic agent, it is becoming clear that monotherapy is inadequate. 

ADCs are an appealing treatment tool given their broad applicability across different tumour types and the possibility to combine with other agents. The most successful example is ado-trastuzumab emtansine (T-DM1). This HER2-directed ADC incorporates the HER2 targeted actions of trastuzumab with the microtubule inhibitor DM1 [91]. T-DM1 was approved by FDA in 2013 after demonstrating a 5.8-month improvement in OS in women with metastatic HER2-postive breast cancer [92]. In 2019, the FDA extended its approval to include adjuvant treatment based on finding from KATHERINE trial that compared T-DM1 with trastuzumab. In this trial, women treated with T-DM1 had a 50% reduced risk of recurrence of cancer or death compared to women treated with trastuzumab alone [93]. Further improvements were made with trastuzumab deruxtecan (T-DXd), an ADC with a membrane permeable cytotoxic payload and tetrapeptide-based cleavable linker [94]. This had a dramatic impact on survival for HER2-positive patients with a HR for disease progression or death of 0.28 (*p* < 0.001) in the DESTINY-Breast03 trial which compared to T-DM1 [95]. More recently, trastuzumab deruxtecan showed encouraging efficacy in patients with HER2 expressing biliary tract cancer [96]. These treatments for HER2-expressing cancers demonstrate successful development of highly effective ADCs in both adjuvant and metastatic setting and across different tumour types. The possible reasons that may have contributed to the failure of Rova-T in pivotal phase III trials in ES-SCLC were discussed in an accompanying editorial [97] and included low drug–antibody ratio (DAR), unacceptably toxic payload and the early cleavage of the linker, releasing the payload into the circulation leading to systemic toxicity. New ADCs targeting DLL-3 with better pharmacologic properties, such as a high DAR, a payload with good therapeutic index and antibody with superior properties, need to be developed.

The failure of Rova-T development strategy in SCLC, on the other hand, also highlighted the dangers of moving directly from promising small phase I studies to large registration phase III studies without confirming the safety and efficacy data in phase II studies or larger expansion cohorts in seamless phase I/II trial. Although this approach has worked for agents targeting oncogenic driver mutation, we should not abandon the principles of good drug development with other agents. The risk is overly optimistic results based on biased phase I trials, with carefully selected relatively good prognosis patients not representative of ‘real world’ experience. Additionally, the over-estimated real benefit of the drug is based on the lack of a control arm.

Combination trials of PARP inhibitors with additional agents have shown promise, with response rates as high as 40% in the olaparib and TMZ trials. Whilst limited by low numbers, the biological foundation for these combinations is being established. This combination is especially attractive in the relapsed setting, given the high prevalence of intracranial disease in SCLC and the demonstrated CNS-control in the available data so far [98]. Ongoing trials will read out in the coming months, and if positive, larger studies should be established to confirm the efficacy of this combination. Despite the negative result of maintenance PARP inhibitor after first line platinum-based chemotherapy [56], adding the combination of PARP and TMZ to the current maintenance ICB may further advance the survival benefit in ES-SCLC. 

One obstacle in SCLC research is a paucity of tissue for comprehensive molecular characterisation, biomarker analysis and a deficiency of clinically relevant model systems both for the study of basic biology of SCLC and the preclinical drug development to inform clinical trials. For example, in the successful IMpower-133 trial of atezolizumab, data on the PD-L1 status of most patients could not be obtained due of insufficient quality of tumour material [99]. Liquid biopsies, including circulating tumour cells (CTCs) and circulating tumour DNA (ctDNA) are increasingly used as surrogates for tumour tissue and have the advantage of being easily obtained serially to inform on the biology of disease progression and acquired chemotherapy and immunotherapy resistance. The genetically chaotic nature of SCLC with multiple chromosomal rearrangements, losses, and gains along with the near universal alterations seen in *TP53* and *RB1* offers great opportunities for the detection of SCLC ctDNA. For example, in a study of serial ctDNA samples from 27 patients, disease-associated mutations were detected in 85% of patients [100]. Although complex and technically challenging, the concerted efforts of preclinical and clinical researchers to conduct orchestrated biomarker-driven trials will hopefully accelerate the advancement in SCLC treatment. 

## 7. Conclusions

In summary, SCLC is a recalcitrant disease and difficult to treat, especially at the time of relapse. Combination therapies with strong scientific rationale in robust clinical trials should continue to be explored. Further research to understand the markers and clinicopathological characteristics that govern response or resistance to targeted and immune therapies represent a fundamental unmet need. 

## Figures and Tables

**Figure 1 ijms-24-08129-f001:**
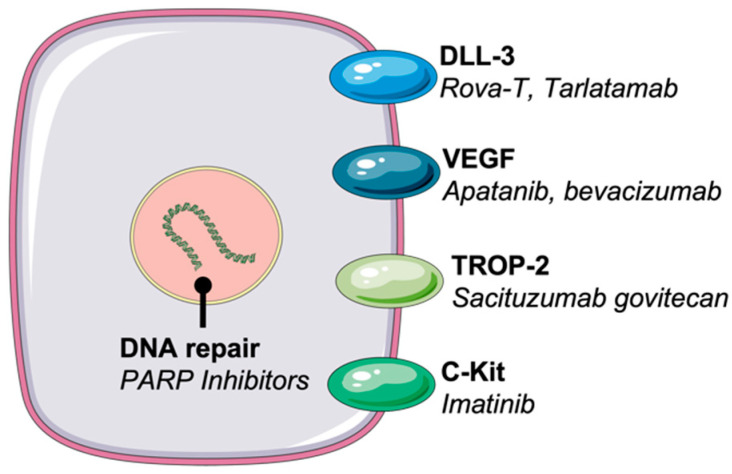
Sites of action of emerging therapeutics for SCLC and examples of agents. The Figure was partly generated using Servier Medical Art, provided by Servier, licensed under a Creative Commons Attribution 3.0 Unported license.

**Table 1 ijms-24-08129-t001:** SCLC Subtype Classification as described by Gay et al. 2021 [9].

Subtype (Key Gene; % of Sample ^1^)	Key Characteristics	Potential Therapeutic Vulnerabilities
SCLC-A (ASCL1; 51%)	Neuroendocrine, epithelial subtype; TTF1 expression	BCL2 inhibitors
SCLC-N (NEUROD1; 23%)	Neuroendocrine, lacks TTF1 expression, cMYC expression	Aurora kinase inhibitors (AURKi)
SCLC-P (POU2F3; 7%)	Less neuroendocrine (NE) expression	PARP inhibitors, antimetabolites, AURKi
SCLC-I (inflamed; 17%) ^2^	Less NE expression, mesenchymal type	Immune checkpoint inhibitors

^1^ Based on the Impower133 dataset; ^2^ SCLC-I expressed no clear transcriptional signature, but numerous immune checkpoints.

**Table 3 ijms-24-08129-t003:** FDA-approved first and second-line therapies for ES-SCLC [32].

First line	Carboplatin or cisplatin, etoposide + atezolizumab or durvalumab
Second line	CAV (cyclophosphamide, doxorubicin, vincristine) Topotecan Lurbinectedin * Nivolumab + Pembrolizumab

* Ongoing approval pending confirmatory study; + Indication withdrawn after initial approval based on CheckMate 451.

**Table 5 ijms-24-08129-t005:** Published anti-angiogenesis trials for recurrent ES-SCLC.

Agents	Study Design	Key Results
Cediranib [69]	Phase II, single-arm N = 25	mPFS 2 mo; mOS 6 mo; No confirmed objective responses
Sorafenib [70]	Phase II, single-arm N = 89	Platinum-sensitive: mOS 6.7 mo; ORR 11% Platinum-resistant: mOS 5.3 mo; ORR 2%
Apatinib [71]	Phase II, single-arm N = 57	mOS 11.2 mo; ORR 14.3%
Anlotinib [72]	Retrospective N = 40	mOS 7.8 mo; ORR 10%
Pazopanib [73]	Phase II, single-arm N = 58	Platinum-sensitive: mOS 8.0 mo; ORR 17.9% Platinum-resistant: mOS 4.0 mo; ORR 5.3%

## Data Availability

Not applicable.

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
