# Peer review of "Small-Cell Lung Cancer—An Update on Targeted and Immunotherapies"

_ijms, 2023, doi:10.3390/ijms24098129_

Round 1

Reviewer 1 Report

The current manuscript is an interesting review on old and new therapies for small-cell lung cancer. It is overall quite thorough and complete, nevertheless some alterations should be made before acceptance for publication:

- The review lacks illustrative images, authors should add, for example, an image illustrating current vs new treatments, and related advantages and limitations; and an image regarding the diseases pathophysiology;

- The meaning of some abbreviations is lacking, for example “PDL1”; while this might be basic knowledge for the authors, defining all abbreviations will help researchers from different fields better understand what is being said;

- A “Conclusion” section is missing, apart from the “Discussion” section.

Reviewer 2 Report

Please add comments concerning the CAPSTONE-1 and Astrum-005 trials in the Immunotherapy paragraph and ed edit the table. 

Please report data concerning Tarlatamb efficacy in the section concerning the new treatments. 

Round 2

Reviewer 2 Report

I think it's situable for the pubblication now